# Activin A Reduces Porcine Granulosa Cells Apoptosis via ERβ-Dependent ROS Modulation

**DOI:** 10.3390/vetsci9120704

**Published:** 2022-12-18

**Authors:** Fang Chen, Xiaoqing Zhu

**Affiliations:** 1Institute of Animal Science, Jiangsu Academy of Agricultural Sciences, Nanjing 210014, China; 2Key laboratory of Crop and Animal Integrated Farming, Ministry of Agriculture, Nanjing 210014, China; 3Science and Technology Industry Development Center, Chongqing Medical and Pharmaceutical College, Chongqing 401331, China

**Keywords:** porcine granulosa cells, activin A, apoptosis, reactive oxygen species, estrogen receptor β (ERβ)

## Abstract

**Simple Summary:**

Activins and inhibins are closely related protein heterodimers with opposing functions in follicular development. The increased circulating follicle-stimulating hormone (FSH) levels and strengthened estrus behavior may result from the immune neutralization of the inhibin bioactivity, which might improve ovarian follicle formation. However, the direct effect of activins, or immunization against inhibin, on the granulosa cells (GCs) functions remains largely unknown. We aimed to examine the effects of activin A (ACT-A) on the function of porcine ovarian GCs. The results showed that ACT-A could suppress ROS accumulation through the upregulation of the expression of estrogen receptor-β (ERβ), thus attenuating apoptosis in the porcine granulosa cells and promoting estradiol synthesis. These results identified a novel protective role of ACT-A in the regulation of the follicle functions, which revealed the mechanism of improvement locally in the ovary caused by immunization against inhibin.

**Abstract:**

Unfavorable conditions compromise animal reproduction by altering the ovarian granulosa cells’ follicular dynamics and normal physiological function (GCs), eventually resulting in oxidative damage and cell apoptosis. Activin is produced in the GCs and plays a vital role in folliculogenesis. This study investigated the effects of activin A (ACT-A) treatment in vitro on the apoptosis of porcine GCs and the underlying molecular mechanism. We found that ACT-A could attenuate the apoptosis of the GCs and enhance the synthesis of estrogen (E2). ACT-A also enhanced FSH-induced estrogen receptor-β (ERβ) expression, inhibiting ERβ aggravated intracellular accumulation of the reactive oxygen species (ROS) and apoptosis. The E2 levels in the culture medium, the mRNA expression pattern of the apoptosis-related genes (*CASPASE 3*, *BCL2*, and *BAX*), steroidogenesis-related gene (*CYP19A1*), and cell viability were analyzed to confirm the results. In summary, this study indicated the protective role of ACT-A in apoptosis by attenuating the ROS accumulation through ERβ. These results aim to enhance the follicular functions and improve animal reproductive performance.

## 1. Introduction

Researchers and practitioners are becoming more concerned about the growing infertility issue in people and animals, which may be brought on by stress and hyperandrogenism-prompted ovarian follicular maldevelopments [1,2,3]. More than 99% of the follicles in mammalian ovaries have atretic degeneration before ovulation [4]. During the past decades, several endocrine manipulating reproductive protocols have been developed to enhance the ovarian functions and improve reproductive performance [5,6], but the current protocols are far from satisfactory. The three interrelated characteristics that impact an animal’s ability to reproduce soundly are a sound-growing ovarian follicle that may create a high-quality egg, a high-quality embryo consequently, and a high-quality corpus luteum [7,8]. By increasing the circulating follicle-stimulating hormone (FSH) concentrations, the bolstering estrus behavior, and enhancing the oocyte and early embryo development competence in dairy cows [9], water buffaloes [10], and pigs [11], a method of immunoneutralization of inhibin bioactivity has been developed in recent decades to effectively stimulate or enhance the granulosa cells and ovarian follicle development. In addition to increasing the conception rate, immunization against inhibin, when combined with the OvSynch protocol, also increased the plasma concentrations of the interferon tau (IFN-tau) in dairy cows around the time of pregnancy recognition, further demonstrating its efficacy in enhancing early embryo development and oocyte maturation [12,13].

Activins and inhibins are closely related protein heterodimers [14], belonging to the transforming growth factor-beta (TGF-β) superfamily, however, these two complexes have opposing functions in follicular development [15,16]. The hormone inhibin is a dimeric glycoprotein which is composed of an α-subunit and either a βA or a βB-subunit (inhibin A and inhibin B, respectively). It is primarily secreted by the gonads and inhibits the pituitary secretion of the FSH through negative feedback regulation [17,18]. Activins are composed of homodimers of β subunits, namely activin A (βA βA), activin AB (βA βB), and activin B (βB βB). In the pituitary, the activin increased the synthesis and secretion of FSH, and this process could be counter regulated by inhibin. The inhibin antagonizes the activin signaling by competitively binding to the activin type 2 receptors (ActRII). It has been hypothesized that, in addition to the stimulation brought on by the increased FSH secretion, the immunoneutralization of the inhibin bioactivity may also directly boost the granulosa or follicular cell function [19]. Additionally, Cai [20] cultivated porcine granulosa cells using an anti-inhibin-subunit antibody and found that, through enhancing activin signaling pathways, the immunoneutralization of inhibin bioactivity allowed the formation of healthy and viable granulosa cells. Currently, the reported effects of activins on the proliferation in the ovary are conflicting. Some groups reported that activin A (ACT-A) played a role in the oocyte maturation and the proliferation of the granulosa cells and pre-antral follicles in mice, as well as increased the FSH receptor (FSHR) expression in vitro. In cattle ovary research, ACT-A was reported to attenuate the apoptosis of the bovine ovarian granulosa cell in the atretic follicles [21]. However, the mechanism underlying the enhanced growth and development of the ovarian follicles by activin remains to be further understood. Others observed the opposite outcome, which hindered human follicular development and lowered the swine granulosa cells in vitro production of estradiol (E2) and progesterone (P4).

Therefore, in this study, we investigated the effect of ACT-A from the perspectives of development and apoptosis on porcine granulosa cells, per se. Our findings revealed a primary protective role of ACT-A, with induced GC survival in an estradiol receptor beta (ERβ) dependent mode.

## 2. Materials and Methods

### 2.1. Granulosa Cell Isolation and Culture

The granulosa cells in the ovaries of prepubertal gilts aged 165–180 days were isolated and grown following the methods used in other investigations. In a nutshell, the follicles with a diameter of 3–6 mm were used to aspirate the granulosa cells using a syringe and sterile needles. The granulosa cells were then separated by centrifuging them for 5 min at 1000× *g*, rinsing them in a sterile F12 medium (Wisent Corporation, Nanjing, China), and resuspending them in the same medium with 10% fetal calf serum (10099141C, Gibco; Shanghai, China) and 1% antibiotic-antimycotic solution (Sigma, St. Louis, MO, USA) at a final density of 10^6^ cells/mL. The cell suspension was then divided into aliquots and placed onto 6-well culture plates made by Nunc International (Naperville, IL, USA; 2 mL/well). The cells were incubated in humidified air with 5% CO_2_ at 37 °C. After 48 h of incubation, the wells were twice rinsed with PBS to remove the single cells and then refilled with 2 mL of brand-new F12 media with 2% fetal calf serum.

### 2.2. Cell Treatment

The unattached cells were eliminated and the cells were treated in a new cell culture media consisting of F12 medium supplemented with 2% FBS and 0.1 µM androstenedione. ACT-A (R&D systems, Minneapolis, MN, USA) was dissolved in the new media at 50 ng/mL. The H_2_O_2_ was diluted to 0.4 mM, as previously reported [22,23]. PHTPP (4-[2-Phenyl-5,7-bis(trifluoromethyl)pyrazolo[1,5-a]-pyrimidin-3-yl] phenol, Sigma, Burlington, MA, USA) was used at the concentration of 10 µM, as previously described [24,25]. The cells were treated for 24 h and harvested for mRNA and protein detection.

### 2.3. Cell Viability Assay

The optical density of the yellow color was measured at 490 nm using a BioTek Eon microtiter plate reader. The GCs were cultured in 96-well plates, and their viability was assessed using the CCK-8 cell viability assay kit (Cell Counting Kit-8; Beyotime Co., Ltd., Shanghai, China), following the manufacturer’s instructions after the heat treatment. The percentage of the absorbance readings compared to the control was used to indicate the cell viability. Three distinct cultures were used in the tests, and three copies of each sample were analyzed.

### 2.4. Measurement of E2 Secretion

According to the manufacturer’s instructions, an ELISA kit (Beijing North Institute of Biological Technology, Beijing, China) was used to quantify E2 in the culture media. The kit’s standard curve covered the concentration range of 0 to 400 pg/mL, and the intra- and inter-assay coefficients of the variation were both less than 10%. Every sample was measured three times.

### 2.5. Gene Expression Analysis

The TRIzol Reagent (74104, Invitrogen, Shanghai, China) was used to separate the total RNA from the grown GCs, and the 1st-Strand cDNA Synthesis Kit (11119ES60, YEASEN, Shanghai, China) was used to reverse-transcribe the total RNA into cDNA, following the manufacturer’s instructions. In porcine granulosa cells, the mRNA expression levels of *β-Actin*, *CYP19A1*, *FSHR*, *BAX*, *CASPASE 3*, *BCL2*, and *ERβ* were quantified using a real-time quantitative polymerase chain reaction (the primer information is shown in Table 1). The PCRs were performed using a One-Step RT-qPCR SYBR Green Kit (11143ES850) on an ABI 7500 (Applied Biosystems; Foster City, CA, USA) with a 20 μL reaction volume. After the real-time qPCR was finished, the ABI 7500 software V.2.0.6 determined the threshold cycle (Ct) values (Applied Biosystems; Foster City, CA, USA). As reported in our earlier work, the 2^−ΔΔCt^ technique was used to quantify the gene expression levels, which were then normalized to the expression levels of the internal housekeeping gene *β-Actin*. The triplicates of each sample were analyzed.

### 2.6. Analysis and Detection of ROS

Using cell-permeant 2′, 7′-dichlorodihydrofluorescein diacetates (H2DCFDA; Beyotime Institute of Biotechnology, Shanghai, China), as previously reported [41], the intracellular ROS levels in the cells following the H_2_O_2_ treatment were measured. In a 24-well plate, the sterile coverslips were inserted in each well before the seeding of the granulosa cells. The cells were treated, as previously indicated, and incubated in H2DCFDA/PBS solutions (1:1000) at 37 °C for 30 min. The coverslips were placed on the glass slides after being thoroughly washed in DPBS (with the cell side laid face down to the glass slide). Finally, a confocal microscope was used to analyze the instantaneousness of the cells (Zeiss LSM700 META). The average pixel intensity of three distinct fields from each experiment was examined for the ROS level analysis (all the cells in each field were examined), and the areas adjacent to the cells that do not fluoresce were designated as the background.

### 2.7. Analytical Oxidative Stress-Associated Parameters

Using the xanthine oxidase technique, the activity of SOD in the granulosa cells was quantified and represented as units per mg of protein. Thiobarbituric acid was used to assess the MDA concentration in the granulosa cells, and the results were expressed as mol per mg of protein. The exact stages were carried out following the instructions included with the kits.

### 2.8. Statistical Analysis

The student’s *t*-test was used to examine the data presented as the mean SD. SPSS Statistics version 25.0 was used to conduct all the statistical analyses (SPSS Inc., Chicago, IL, USA). The cutoff for the statistical significance was *p* < 0.05.

## 3. Results

### 3.1. ACT-A Enhances the Expression of FSHR and ERβ and Significantly Increases Granulosa Cells’ Sensitivity (GCs) Sensitivity to FSH Treatment

As shown in Figure 1A, ACT-A increased the gene expression of *FSHR*. As a result, we included FSH in the following treatment to explore the effect of ACT-A on the sensitivity of the GCs to the FSH. It was found that co-treatment of ACT-A and FSH dramatically enhanced the secretion of estrogen (E2, Figure 1B) and the gene expression of *CYP19A1* (Figure 1C), suggesting that ACT-A demonstrates a synergetic effect in FSH-induced follicle development. As reported, *ERβ* expression contributes to E2 synthesis and plays a role in FSH-mediated follicle development. Therefore, the impact of ACT-A and co-treatment of ACT-A and FSH on *ERβ* abundance was investigated in this study. As shown in Figure 1D, the *ERβ* expression level was significantly increased under both conditions, suggesting a potential role of *ERβ* underlying ACT-A on the GC’s growth and survival.

### 3.2. ACT-A Attenuates Apoptosis of GCs

As observed in the CCK-8 assay (Figure 2A), ACT-A treatment inhibits the GC’s apoptosis. So, using the real-time PCR, we determined the prevalence of the genes *BAX*, *BCL2*, and *CASPASE 3* associated with apoptosis. According to Figure 2B, ACT-A inhibits the apoptosis of the GCs, as shown by the significantly reduced relative expression of *BAX* and *CASPASE*
*3*, and the elevated expression of *BCL2*.

### 3.3. ACT-A Mediates GCs Apoptosis via Modulating ERβ Expression

Based on the findings above, we hypothesized that ERβ modulation could be the underlying mechanism for ACT-A effect on the GC’s apoptosis. As a result, we treated the GCs with the selective ERβ inhibitor PHTPP (10 μM), in combination with ACT-A, and then examined the gene expression of *BAX*, *BCL2*, and *CASPASE 3*. Firstly, the decreased *CYP19A1* expression was used to confirm the effect of PHTPP on the E2 synthesis (data not shown). Next, as shown in Figure 3C, the *BAX* and *CASPASE 3* expressions were significantly increased, while the *BCL2* expression was downregulated with both the PHTPP treatment alone and the co-treatment of PHTPP and ACT-A, indicating an induction effect of ERβ inhibition on the cell apoptosis. Meantime, it is remarkable that the PHTPP also inhibits the *ERβ* expression in Figure 3B, further implying an important role of the *ERβ* expression in the observed ACT-A effect. Finally, the CCK-8 assay was utilized in the following to validate this finding. As shown in Figure 3A, the result demonstrated that the PHTPP treatment promotes the GC’s apoptosis and thus verified the hypothesis that ACT-A mediates the GC’s apoptosis via the modulation ERβ expression. 

### 3.4. ACT-A Mediates Intracellular ROS Levels in GCs via ERβ

To further investigate the underlying mechanism by which ACT-A and ERβ regulate the GC’s apoptosis process, we drew attention to the cellular ROS level detection. As shown in Figure 4A, the ACT-A treatment could decrease the intracellular ROS levels of the granulosa cells, which might contribute to the inhibitory effect of ACT-A on apoptosis. Meanwhile, it was observed that the co-treatment of PHTPP with ACT-A attenuated the impact of ACT-A on ROS, further implying the role of ERβ in this process. Lipid peroxidation (MDA) detection (Figure 4B) and Superoxide Dismutase (SOD) measurement (Figure 4C) were thus performed, and the data indicated that ACT- A reduces ROS-mediated apoptosis through ERβ.

## 4. Discussion

In this work, we demonstrated how ACT-A changed the E2 production and the cell death of granulosa cells. Substantial modulation of ROS by ERβ expression was found to contribute to these reactions. These results effectively combine the currently known effects of activin as a local factor on follicular growth with the impact of inhibin vaccination on the granulosa cell proliferation and steroid hormone release to enhance the follicle development.

GCs are steroidogenic cells surrounding the oocyte, which play an essential role in follicular development, oocyte maturation, and the subsequent embryo implantation [26]. The maintenance of the GCs contributes to normal follicular growth and, in particular, plays an important role in deciding the fate of the follicles. The apoptotic GCs may cause follicle development disturbance, poor quality of oocytes, and induce low reproductive performance [27]. As reported, multiple apoptotic signaling molecules in the GCs, such as hormones, growth factors, death ligand-receptor system, and Bcl-2 family members, affect each other, and activate Caspase 3 and the subsequent DNA fragmentation, and result in the GC’s apoptosis [28].

From the earliest stages of follicle development, the granulosa cells begin to produce activin, and as follicle growth progresses, inhibin/follistatin, which inhibits activin’s actions, takes over in the granulosa cells [29]. The granulosa cells of dominant or prominent ovarian follicles are primarily responsible for the inhibin secretion [30]. Through a negative feedback loop [31], inhibin suppresses pituitary follicle-stimulating hormone (FSH) release. It limits the growth of subordinate follicles through the para/autocrine regulatory pathways, which adversely control ovarian follicular development [32]. Inhibin is known to act locally in the ovary, the most clearly defined paracrine function being to antagonize the effect of activin [33]. By passive or active immunization against the inhibin subunit peptide, the immuno-neutralization of inhibin’s ovarian follicle suppression activity improves the follicle development and hormone secretion capacity, significantly boosting ovulation rates and reproductive performance in various animal models.

According to reports, activin induces the expression of FSH-R and LH receptors, while also increasing the activity of the FSH-induced aromatase. In this study, after 24 h of culture, ACT-A was added to the culture medium. Compared with the cultured control, ACT-A promoted E2 secretion and caused an increase in *FSHR* mRNA levels and, as a result, increased GC sensitivity to FSH, which led to an increased *CYP19A1* expression. Additionally, we discovered that ACT-A stimulates *ERβ* expression, and ERβ inhibition results in a deficit in E2 production. Our results align with the earlier research that suggests the GCs play a fundamental role in forming the ovarian follicles stimulated by FSH [21]. According to in vitro studies, the presence of ERβ in GCs is necessary to produce a subset of FSH-induced genes, including *CYP19A1* [21]. In addition, previous studies conducted on ERβ-null mice reported that the knockout of ERβ significantly reduced the levels of FSH-induced estrogen synthesis [34,35]. In the meantime, the expression of ERβ is regulated by FSH through the PI3K/AKT pathway [35]. Our findings demonstrated that ACT-A increased the expression of *ERβ* caused by FSH. The underlying regulatory mechanism of the FSH, ERβ, and CYP19A1 network might explain the phenomenon that the elevation of E2 induced by ACT-A depends on FSH’s presence [36].

The cellular redox status is crucial to cell survival, growth, and death. The accelerated metabolic rates and the cumulative accumulation of the reactive oxygen species (ROS) are linked to higher demands for energy and nutrients throughout the reproductive process. Unfavorable environmental conditions, such as bacterial infection and heat stress, are well known to promote the accumulation of ROS [22,37]. Although ROS are by-products of aerobic metabolism that are naturally occurring, the excessive ROS production causes oxidative stress and cellular damage. In GCs, the excessive intracellular ROS could cause a series of damage, such as disruptive apoptosis, altered cell proliferation, and disordered E2 synthesis [23,38]. The death of GCs during follicular atresia, which may result in certain anovulatory illnesses, such as premature ovarian failure, is strongly supported by the data that oxidative stress plays a vital role in the process [39].

In this study, we demonstrated that ACT-A attenuated the GC’s apoptosis via mediating ROS production, and induced *ERβ* expression in the GCs. As observed in the rat ovary follicular growth and atresia development, the *ERβ* expression level decreases along with the apoptosis increasing in the follicular granulosa cells [40]. Moreover, ERβ was found to interact with vigilin to protect the ovarian granulosa cell-like human granulosa cells from the palmitic acid-induced apoptosis [41]. We thus hypothesized and validated that ERβ plays an essential role in the ACT-A action mode. Our data showed for the first time that the inhibition of ERβ significantly enhanced the intracellular ROS level of the GCs and resulted in a remarkable increase in cell apoptosis. 

According to available research on ERβ, it may promote invasion, adhesion, inflammatory body activity, proliferation, and inflammatory signals of the ectopic lesions, while inhibiting apoptosis [42,43], which verifies the unveiled protective role of ERβ in our study. To prevent TNF-α induced apoptosis, it is well known that ERβ interacts with the cellular apoptotic machinery in the cytoplasm [44,45]. Meanwhile, synchronized changes in ERβ expression and ROS induction were detected in the human granulosa cells [46]. It is indicated in the human granulosa cell line KGN study that both the ERβ expression and the ROS-ASK1-JNK axis take part in Bisphenol AF-induced cell apoptosis [47]. A working model was also further proposed for the protecting action of ERβ against seminoma, in which ERβ regulated the gene expression of *SIRT3*, a major mitochondria nicotine adenine dinucleotide (NAD)+-dependent deacetylase, and resulted in ROS level reduction [46]. Therefore, our study proves that ACT-A could mediate the GC’s apoptosis in ERβ-dependent mode, and gene expression in mitochondrial adaptative responses to stress might be a potential mechanism for further exploration.

## 5. Conclusions

According to our results, ACT-A can suppress the ROS accumulation through the upregulation of the expression of ERβ, thus attenuating apoptosis in porcine granulosa cells. These results identified a novel protective role of ACT-A in the regulation of follicle functions, which revealed the mechanism of improvement locally in the ovary caused by the immunization against inhibin. This study also provides proof of principle for enhancing follicular functions and improving animal reproductive performance.

## Figures and Tables

**Figure 1 vetsci-09-00704-f001:**
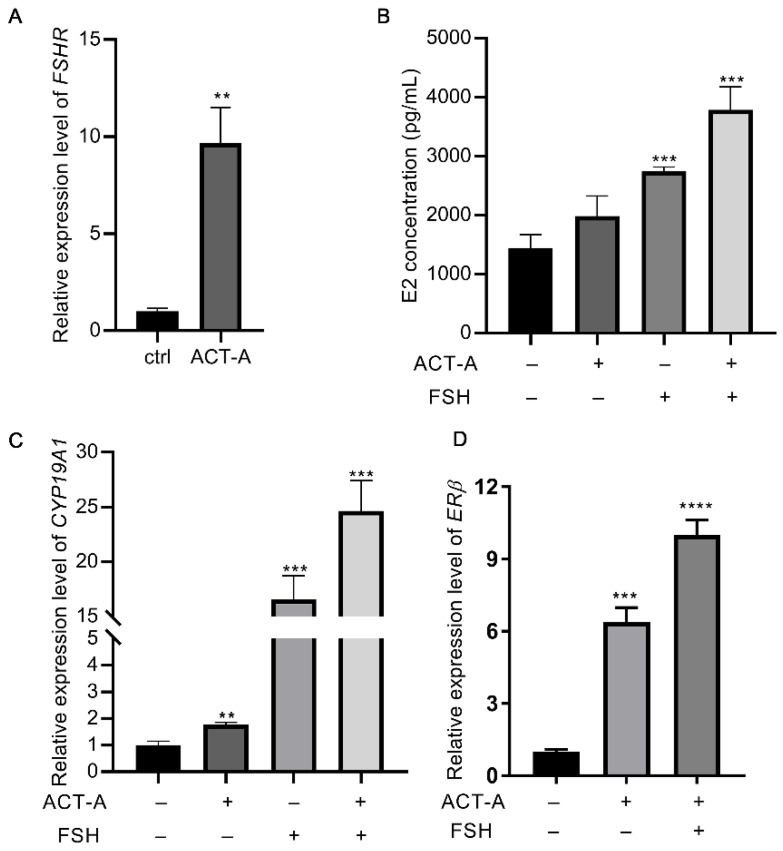
Effect of ACT-A on *ERβ* and *FSHR* expression and GC sensitivity to FSH therapy. qRT-PCR was used to evaluate the gene expression level of *FSHR* (**A**), *CYP19A1* (**C**), *ERβ* (**D**), and E2 concentration levels in GCs (**B**). The ctrl means control (untreated group). ** *p* < 0.01, *** *p* < 0.001, **** *p* < 0.0001.

**Figure 2 vetsci-09-00704-f002:**
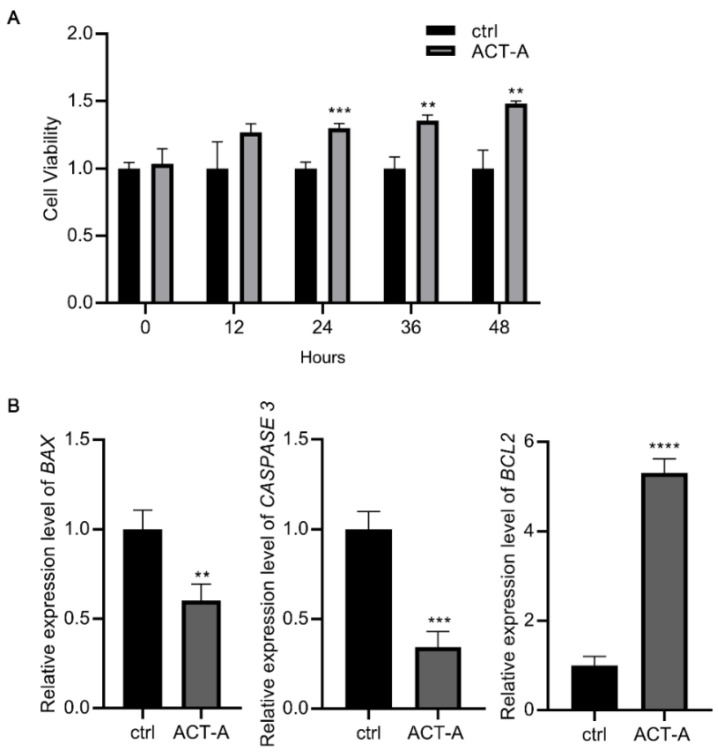
Effect of ACT-A on apoptosis of GCs. (**A**). The CCK-8 test was used to determine how ACT-A affected the apoptosis of GCs. (**B**). Expression of apoptosis-related genes *BAX*, *CASPASE 3*, and *BCL2* was assessed using qRT-PCR. ** *p* < 0.01, *** *p* < 0.001, **** *p* < 0.0001.

**Figure 3 vetsci-09-00704-f003:**
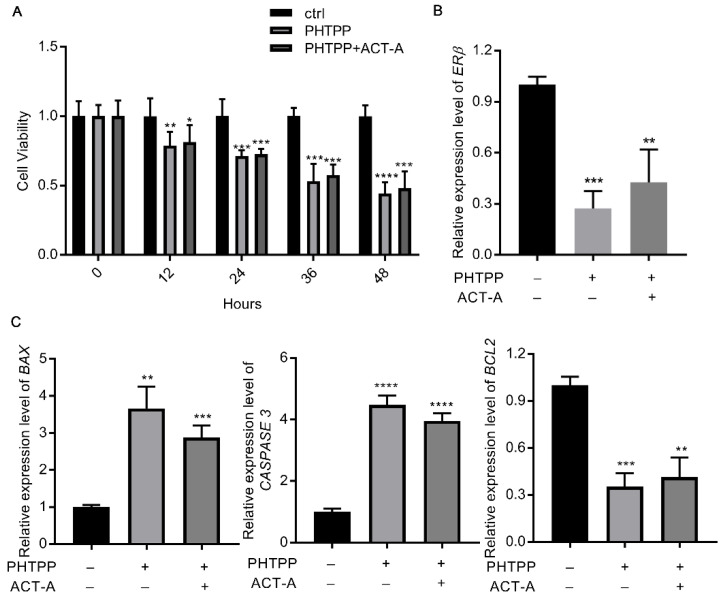
ACT-A mediates GCs apoptosis via modulation ERβ expression. (**A**). The CCK-8 test was used to determine how PHTPP affected the apoptosis of GCs. B. C. *Erβ.* (**B**). *BAX*, *CASPASE 3*, and *BCL2* (**C**). The gene expression levels were assessed by qRT-PCR. * *p* < 0.05, ** *p* < 0.01, *** *p* < 0.001, **** *p* < 0.0001.

**Figure 4 vetsci-09-00704-f004:**
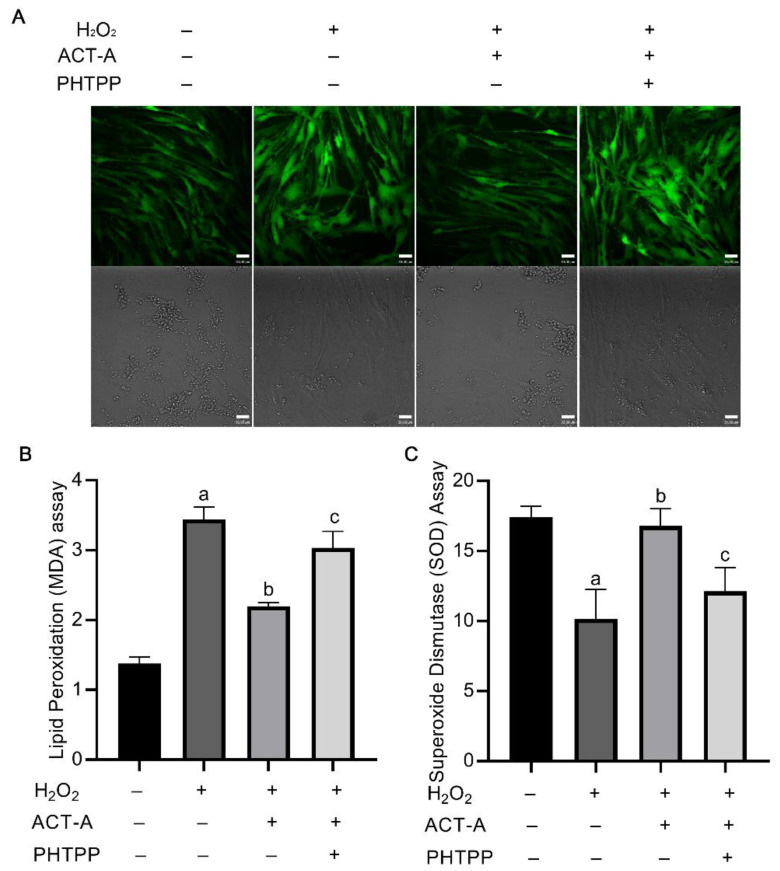
ACT-A mediates intracellular ROS level of GCs in ERβ-dependent mode. (**A**). Measurement of cellular ROS level by treatment of ACT-A and PHTPP. (**B**). Lipid peroxidation assay was performed to detect MDA changes. (**C**). Superoxide dismutase assay was performed to detect SOD changes. “a” means a significant change compared with the control (untreated group), “b” means a substantial change to H_2_O_2_ treatment, and “c” means a considerable change to co-treatment of H_2_O_2_ and ACT-A.

**Table 1 vetsci-09-00704-t001:** Primers utilized in this investigation.

Gene	Primer Sequences (5′—3′)	Length
*β-Actin*	F: CTTCCTGGGCATGGAGTCC	201 bp
	R: GGCGCGATGATCTTGATCTTC	
*CYP19A1*	F: GGTCACAACAAGACAGGA	168 bp
	R: AACCAAGAGAAGAAAGCC	
*FSHR*	F: GCCCAGAACTAAAACACAATG	107 bp
	R: TATAGACAAGTAACCGTCAGC	
*BAX*	F: AATTGGCTTGGTCTGTAT	104 bp
	R: CGGTCGTGATGGTATGTG	
*BCL2*	F: CATGCGTATTTATATTTG	112 bp
	R: CTCTGCTGCTTGCTGCTA	
*CASPASE 3*	F: ATGTCAGGCTAGTCTCTC	124 bp
	R: TGGTATGTAACTTGGGGA	
*ERβ*	F: TATCTCCTCCCAGCAGCAGTCT	153 bp
	F: TATCTCCTCCCAGCAGCAGTCT	

## Data Availability

The corresponding author may provide the data used in this research upon request.

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
