# Peer review of "Activin A Reduces Porcine Granulosa Cells Apoptosis via ERβ-Dependent ROS Modulation"

_vetsci, 2022, doi:10.3390/vetsci9120704_

Round 1

Reviewer 1 Report

This manuscript provides an in-depth investigation of the protective effect of ACT-A on GCs, which was found to inhibit ROS accumulation and attenuate apoptosis of GCs via ERβ. The results of this study are of great significance for both the application of ACT-A in basic research and the exploitation of ACT-A in practical production. However, the manuscript currently has the following points for refinement and improvement.

a)       In result 1, after finding that ACT-A with FSH enhanced E2 secretion, the authors found that ACT-A treatment alone enhanced ERβ expression levels. It is suggested that a combined ACT-A and FSH treatment be added here, which would better explain the enhanced effect of the combined treatment.

b)      The treatment procedure (concentration selection, time selection) for ERβ inhibitor PHTPP needs to be described.

c)       ACT-A is shown in Figure 1D to significantly upregulate ERβ mRNA upregulation. PHTPP was used for validation in all subsequent experiments. However, it is generally accepted that PHTPP acts by inhibiting E2-stimulated ERβ activity. Therefore, the authors need to consider which pathway ACT-A exerts its protective effect through upregulation of ERβ expression levels or enhancement of ERβ activity. In addition, it needs to be considered whether the apoptosis due to ERβ inhibition is the result of reduced expression levels or diminished activity. Additional Erβ siRNA transfection or other experiments are suggested to elucidate this issue.

d)      Detection of apoptosis in GCs after combined treatment with ACT-A and PHTPP needs to be added.

e)       It is suggested to supplement the results of the detection of ER, BAX, Caspase 3, and Bcl-2 protein levels by Western Blot.

f)       A list of abbreviations needs to be added to the manuscript.

g)      It is suggested that the 0h group in Figure 2A and Figure 3A also be labeled “ns” like the 12h group. Or all “ns” are unmarked.

h)      The way in which gene names are written needs to be modified. The generally accepted way of writing the names of livestock genes is to capitalize all letters and italicize them.

Author Response

Dear Reviewer,

Thank you very much for your advice. We have revised the manuscript and would like to re-submit it for your consideration. We have addressed most of the comments and the amendments are tracked in the revised manuscript. Point by point responses to your comments are listed in the following:

Comment a): “In result 1, after finding that ACT-A with FSH enhanced E2 secretion, the authors found that ACT-A treatment alone enhanced ERβ expression levels. It is suggested that a combined ACT-A and FSH treatment be added here, which would better explain the enhanced effect of the combined treatment.”

Answer:We added co-treatment of ACT-A and FSH for ERβ expression detection and the result was displayed in Fig. 1D (manuscript revised version, attached below). ERβ expression level was significantly increased under both conditions, which was in accordance with the pattern of CYP19A1 expression and E2 secretion and further confirmed the enhanced effect of ACT-A and FSH combined treatment.

Comment b):“The treatment procedure (concentration selection, time selection) for ERβ inhibitor PHTPP needs to be described.”

Answer:The concentration and time of PHTPP treatment were determined from previous studies. In specific, PHTPP (sigma, USA) was diluted at the concentration of 10 µM in F12 medium supplemented with 2% FBS and 0.1µM androstenedione and cells were treated for 24h before harvested for the following detections. The information has been updated in the “Materials and Methods” section (2.2 Cell treatment) and corresponding references were added (#49 & #50).

Comment c):“… PHTPP was used for validation in all subsequent experiments. However, it is generally accepted that PHTPP acts by inhibiting E2-stimulated ERβ activity. Therefore, the authors need to consider which pathway ACT-A exerts its protective effect through upregulation of ERβ expression levels or enhancement of ERβ activity. In addition, it needs to be considered whether the apoptosis due to ERβ inhibition is the result of reduced expression levels or diminished activity. Additional ERβ siRNA transfection or other experiments are suggested to elucidate this issue.”

Answer:We highly appreciated the suggestion above which enlightened us for seeking the mechanism underlying. To elucidate this issue, we carried out experiment to determine the expression level of ERβ in both conditions of PHTPP alone and co-treatment of PHTPP and ACT-A. As shown in Fig. 3B (manuscript revised version, attached below), PHTPP exerts inhibitory effect on ERβ transcription and ACT-A treatment could no longer rescue the expression level with co-treatment with PHTPP. Therefore, we confirmed an important role of ERβ gene expression in observed ACT-A effect. To explain the decreased effect of PHTPP on ERβ expression, we believe that inhibition of ERβ activity by PHTPP could cause changes in expression levels of downstream target genes (such as Cyp19a1, Cyp11a1, Gata 4, etc.) and their reduction may act in a feedback loop to decrease ERβ transcription level. The closed loop formed by PHTPP treatment would lead to apoptosis of granulosa cells. We updated the corresponding sections (Section 3.3) and addressed the mechanism as suggested.

Comment d): “Detection of apoptosis in GCs after combined treatment with ACT-A and PHTPP needs to be added.

Answer:As suggested, we detected the GCs apoptosis with co-treatment of ACT-A and PHTPP and demonstrated the result in Fig. 3A (manuscript revised version, attached below). As shown, both PHTPP treatment alone and combination treatment of ACT-A and PHTPP promoted GCs apoptosis and thus verified the hypothesis that ACT-A mediates GCs apoptosis via modulation ERβ expression.

Comment e): “It is suggested to supplement the results of the detection of ER, BAX, Caspase 3 and Bcl-2 protein levels by Western blot.

Answer:Since the commercial ERβ antibody for Western blot detection did not work in our cellular system, ERβ protein level could not be analyzed in this study. Next, we tried our best to detect the protein levels of BAX, Caspase 3 and Bcl-2. However, BAX antibody was the only one available and obtainable in the short revision period for us during COVID pandemic (2022/12/1-12/10). Thus, we here presented and analyzed the protein level changes of BAX with treatment as requested. In agreement with the data from transcription detection, ACT-A significantly decreased protein level of BAX, whereas both PHTPP treatment alone and combined treatment of ACT-A and PHTPP significantly increased its expression.

Comment f): “A list of abbreviations needs to be added to the manuscript.

Answer:Abbreviation section has been added above the section of “References” in the revised manuscript.

Comment g): “It is suggested that the 0h group in Figure 2A and Figure 3A also be labeled ‘ns’ like the 12h group. Or all ‘ns’ are unmarked.

Answer: Corresponding changes have been made in both Figure 2A and 3A (revised version). All “ns” are unmarked.

Comment h): “The way in which gene names are written needs to be modified. The generally accepted way of writing the names of livestock genes is to capitalize all letters and italicize them.

Answer:Corresponding changes have been made throughout the whole revised manuscript. All the gene names were capitalized and italicized.

Finally, we sincerely value your precious input to our manuscript. We will continue to do our best and are willing to listen to any suggestions you may have.

We are thankful for the suggestions concerning the mechanism and apoptosis which are beneficial for further development of our project. And your advice for the display of the manuscript contributed to our growth as young scientists. Thanks!

Reviewer 2 Report

I would like to ask some questions to the authors:

How many ovaries of prepuberal gilts were used?

Do authors consider it could be a novel method to improve sow reproductive performance?

Considering that this study provides proof of principle for improving follicular functions and improving reproductive performance in pigs, are the authors developing new studies?

Author Response

Dear Reviewer,

Thank you very much for your advice. We are extremely grateful to your interest in this study. Point by point responses to your comments/questions are listed in the following:

Question a): “How many ovaries of prepuberal gilts were used?”

Answer:In this study we used about 200~400 ovaries. However, since several studies are being carried out at the same time in our lab, an exact number would be difficult to be given.

Question b): “Do authors consider it could be a novel method to improve sow reproductive performance?”

Answer:We have carried out studies to develop a novel method to improve the reproductive performance of livestock by immunization against inhibin and achieved significant improvement.

Question c): Considering that this study provides proof of principle for improving follicular functions and improving reproductive performance in pigs, are the authors developing new studies?

Answer:This work is the extension of a previous work carried out by the authors. But at present, inhibin antigen is only used in research studies, not licensed to commercial usage on animals yet. We are looking forward to cooperation with scientists and technicians from both fundamental and applied research in the near future.

We are thankful for the questions concerning the development of our study. And your generosity contributed to our growth as young scientists. Thanks!

Round 2

Reviewer 1 Report

There are still some minor problems with the manuscript. For example, the "P" for prominence is sometimes capitalized (line 153) and sometimes lower-cased (line 170). Another example is the inconsistency of the modified gene names in 28 lines. It is recommended to double-check and revise the full manuscript.

In addition, although only the BAX protein expression changes were detected by WB, it is still necessary to provide the original images of the three replicates, even though this result may not be added to the manuscript or may only appear in the supporting documentation.

The manuscript will be able to be accepted after careful revision.
